# Void Detection of Airport Concrete Pavement Slabs Based on Vibration Response Under Moving Load

**DOI:** 10.3390/s25154703

**Published:** 2025-07-30

**Authors:** Xiang Wang, Ziliang Ma, Xing Hu, Xinyuan Cao, Qiao Dong

**Affiliations:** Department of Roadway Engineering, Southeast University, Nanjing 210000, China; wxiangseu@163.com (X.W.); 220233320@seu.edu.cn (Z.M.); cxyseu@foxmail.com (X.C.)

**Keywords:** moving load, airport pavement, vibration response, void detection, structural health monitoring

## Abstract

This study proposes a vibration-based approach for detecting and quantifying sub-slab corner voids in airport cement concrete pavement. Scaled down slab models were constructed and subjected to controlled moving load simulations. Acceleration signals were collected and analyzed to extract time–frequency domain features, including power spectral density (PSD), skewness, and frequency center. A finite element model incorporating contact and nonlinear constitutive relationships was established to simulate structural response under different void conditions. Based on the simulated dataset, a random forest (RF) model was developed to estimate void size using selected spectral energy indicators and geometric parameters. The results revealed that the RF model achieved strong predictive performance, with a high correlation between key features and void characteristics. This work demonstrates the feasibility of integrating simulation analysis, signal feature extraction, and machine learning to support intelligent diagnostics of concrete pavement health.

## 1. Introduction

As a critical component of aviation infrastructure, airport cement concrete pavements are essential for ensuring the safe takeoff and landing of various types of aircraft, thereby directly influencing the operational safety and efficiency of the air transportation system [1,2,3]. Over extended service periods, these pavement slabs are susceptible to the formation of localized voids beneath the slab base, primarily due to subgrade settlement, base layer degradation, or construction deficiencies [4,5]. Voids in rigid pavement slabs are most prone to develop at the corners due to repeated wheel loading, temperature-induced curling, and erosion of subgrade support. These locations exhibit stress concentrations and uplift tendencies, making them vulnerable to early-stage damage such as debonding and delamination. Such voids are most commonly observed at slab corners, where structural support is inherently limited. The presence of voids beneath the slabs significantly compromises the overall structural stiffness and alters boundary constraint conditions, which in turn accelerates load-induced deterioration and the propagation of pavement distresses [6,7]. As a result, the rapid and accurate detection of sub-slab voids has become a critical technical requirement for the effective maintenance and management of airport pavements [8,9,10].

Currently, void detection techniques primarily rely on deflection response analysis, with the Falling Weight Deflectometer (FWD) being the most widely used tool in engineering practice [11,12]. However, these methods are generally point-based, offering limited spatial resolution and reduced sensitivity to localized damage, particularly at slab corners, where structural vulnerabilities are most pronounced [13,14]. Moreover, they involve complex loading and rebound procedures, which can result in low operational efficiency and potential disruptions to pavement usage. In recent years, structural health monitoring (SHM) approaches based on vibration response analysis have attracted growing interest [15,16]. By analyzing variations in modal parameters and frequency-domain features under external excitation, these methods provide a promising alternative, offering advantages such as real-time monitoring capability and non-contact measurement [17,18].

Previous studies have shown that the presence of voids beneath pavement slabs markedly influences their vibration response characteristics, particularly in terms of shifts in dominant frequency, changes in energy distribution, and alterations in mode shapes [19,20]. Accordingly, developing a void detection framework based on vibration response data—incorporating feature extraction and analytical modeling—offers significant potential for the rapid and quantitative identification of corner voids in pavement structures [21,22]. However, most existing research is limited to idealized simulation settings or static point loading conditions, lacking systematic investigation of structural behavior under realistic loading scenarios. In particular, the correlation between multi-sensor dynamic responses and void characteristics under moving loads remains insufficiently studied [12,23,24].

In recent years, data-driven approaches—such as Support Vector Machines (SVM), Backpropagation (BP) Neural Networks, and deep learning classifiers—have been explored for pavement distress classification and void identification. While these methods show potential for pattern recognition, they often require large labeled datasets, and their interpretability remains limited. Some studies have also utilized modal parameter shifts or deflection ratios for regression-based void estimation. However, most of these rely on simplified or static loading conditions, which do not capture the dynamic nature of real aircraft-induced responses.

In contrast, this study adopts a regression-based detection framework under moving load conditions, integrating localized vibration features with structural parameters. The proposed method aims to bridge the gap between dynamic physical behavior and feature-driven modeling, offering a complementary perspective to classification-based or global modal approaches.

This study aims to develop a multi-scale analysis methodology that integrates experimental validation and numerical simulation for detecting sub-slab voids in cement concrete pavements based on vibration response characteristics. To this end, a 1:10 scaled indoor model test was conducted to replicate the vibration behavior of pavement slabs under impact loading, thereby examining the influence of varying void sizes on structural dynamic responses. Subsequently, a three-dimensional finite element model was constructed in ABAQUS to simulate the vibration responses of slab corners under moving load conditions. From these simulations, time-domain, frequency-domain, and power spectral features were systematically extracted. Finally, by establishing correlations between structural parameters and vibration-sensitive indicators, a detection model tailored to airport operational scenarios was developed. This work lays a technical foundation for the accurate monitoring and intelligent assessment of sub-slab voids in airport pavement systems.

While the scaled laboratory experiment cannot fully replicate the long-term aging effects, structural variations, and material heterogeneity present in full-scale airport pavements, it provides a controlled environment to isolate the influence of void size on vibration response. To enhance the realism of the findings, a high-fidelity 3D finite element model was also developed to simulate real-world loading and boundary conditions.

## 2. Methods

To systematically examine the impact of sub-slab voids on the vibration response characteristics of airport concrete pavements and to develop a detection method suitable for practical engineering applications, this study proposes a multi-scale analysis framework that integrates experimental and numerical approaches. The framework comprises two key components: (1) a scaled laboratory experiment designed to physically capture and characterize vibration patterns under varying void conditions, and (2) a finite element simulation under moving load conditions aimed at quantifying the relationship between critical dynamic features and the extent of voiding.

### 2.1. Scaled Model Experiment

#### 2.1.1. Material and Sample Preparation

A 1:10 scaled model of a cement concrete pavement was constructed, comprising a surface layer and a base layer with dimensions of 0.4 m × 0.4 m × 0.04 m and 0.4 m × 0.4 m × 0.03 m, respectively. The surface layer was cast using C50-grade mortar, while the base layer was formed with C30-grade mortar, as illustrated in Figure 1a. A manually compacted soil bed was used as the foundation to provide structural support and ensure boundary stability.

To closely replicate the service conditions of actual airport concrete pavement structures, an epoxy adhesive was applied at the interface between the surface and base layers to enhance interlayer bonding as shown in Figure 1b. Following the bonding process, the specimen was allowed to cure for 24 h to ensure adequate adhesive strength prior to subsequent loading tests. The specimen was then moist-cured for 28 days prior to testing.

Recognizing that slab corners are the most vulnerable regions to void-induced damage, this study focuses specifically on void scenarios located at slab corners. To simulate these conditions, a handheld angle grinder was used to locally abrade the surface of the base layer, thereby intentionally disrupting the bond between layers and generating artificial voids as shown in Figure 1c,d. The abrasion depth was controlled at 1 mm to represent the shallow delamination commonly observed beneath slab corners in field conditions.

The void was shaped as an equilateral triangle to provide a simplified representation of localized interlayer separation. Four void sizes were considered: 0 cm (no void), 5 cm, 10 cm, and 15 cm, corresponding to increasing degrees of void development. The detailed configuration of each void case is shown in Figure 2.

#### 2.1.2. Sensor Layout and Data Collection

Based on prior research, INV982X piezoelectric accelerometers—developed by the Beijing Institute of Vibration and Noise—were selected for vibration signal acquisition. These sensors feature a measurement range exceeding 25 g, a resolution finer than 0.0005 g, and a frequency response spanning 0.5 Hz to 4 kHz, making them well-suited for capturing dynamic responses associated with sub-slab voids (specifications provided in Table 1).

To capture spatial variation in dynamic response, four accelerometers (Sensors #1–#4) were arranged along the diagonal of the pavement slab, numbered sequentially from right to left. Sensors #2 and #4 were positioned symmetrically relative to the slab corners, maintaining equal distances from the edges, to enable comparison of responses at mirrored positions. The artificial void was introduced beneath the region encompassing Sensors #1 through #3, as shown in Figure 3a. Installation was carried out by embedding M5 bolts into pre-drilled holes on the slab surface. These bolts were secured using AB adhesive and allowed to cure for 24 h to ensure mounting stability (Figure 3b). The accelerometers were then affixed to the bolts via threaded connections, completing the measurement array.

#### 2.1.3. Impact Loading and Signal Acquisition

To simulate external excitation, an INV931X IEPE impact hammer was employed to apply transient loads at the center of the slab surface. The initial impact force was controlled at approximately 10 kN, and all strikes were performed by the same operator to ensure consistency (Figure 4). The loading protocol began with a baseline test under the no-void condition, followed by sequential tests corresponding to void sizes of 5 cm, 10 cm, and 15 cm. For each void scenario, the same loading procedure was repeated to obtain consistent vibration responses, resulting in a comprehensive dataset for the evaluation of the effects of void severity on dynamic behavior.

### 2.2. Finite Element Simulation Under Moving Loads

#### 2.2.1. Model Development and Void Implementation

A dynamic finite element model of airport concrete pavement was developed to simulate slab responses under void conditions, as shown in Figure 5. The model consists of a surface layer composed of nine slabs, each measuring 5 m × 5 m × 0.4 m, underlain by a 0.4 m thick, continuously graded crushed stone base and a 15 m deep subgrade. The subgrade depth was determined based on Saint-Venant’s principle to minimize boundary reflections and eliminate wave interference in the simulation.

All structural layers were modeled using C3D8R elements, with local mesh refinement applied to the central slab to capture stress concentrations. Mesh sizes were set at 0.1 m for the surface layer, 0.2 m for the base layer, and 0.5 m for the subgrade, providing a balance between computational efficiency and solution accuracy. Interlayer contact was defined using a penalty-based friction model with a coefficient of friction of 0.6, while “hard contact” was specified in the normal direction to realistically simulate slip and separation behavior.

To replicate actual load transfer mechanisms, twelve dowel bars (26 mm in diameter, spaced at 400 mm with 400 mm embedment length) were embedded across slab joints in accordance with the Chinese specification MH/T 5004-2010 [25]. The dowel bars were coupled to the surrounding concrete using the “Embedded Region” constraint in ABAQUS, enabling accurate simulation of shear force and bending moment transfer. Material properties—including elastic modulus, Poisson’s ratio, and density—for each structural layer are listed in Table 2 and Table 3, based on design specifications and practical engineering data.

Considering that corner voids typically propagate inward along a diagonal direction, voids were idealized as isosceles right triangles with legs aligned parallel to the slab edges. The leg length served as a quantitative indicator of void extent as shown in Table 3. The voids were modeled as geometric cutouts between the surface and base layers as shown in Figure 6. The light blue grid represents the meshed pavement structure, while the yellow circles denote the selected nodes used for response extraction. Contact within the void region was defined as “free separation” (i.e., frictionless and non-load transferring), simulating interface failure due to delamination.

Although the introduction of contact definitions and a hyperelastic tire model increases computational effort, these settings were necessary to accurately simulate the nonlinear dynamic behavior of slab–void interaction under moving loads.

In total, 16 simulation cases were conducted, each requiring approximately 4–6 h on a 24-core workstation. These simulations were used exclusively for offline data generation, including feature extraction and model training. The final prediction model operates independently of the FE simulations and supports real-time application without additional computation.

#### 2.2.2. Sensor Layout and Numbering Scheme

To capture the dynamic response in the void-affected corner region, a series of measurement points were positioned along the bottom surface of the slab, 10 cm from the lateral edge. Sensors were spaced at 20 cm intervals, beginning 20 cm from the right edge and extending 480 cm toward the left, resulting in 24 measurement points distributed over a total length of 460 cm. This configuration effectively covered the corner area and its adjacent regions on the 5 m × 5 m slab.

The measurement points were sequentially labeled P0 to P23 from right to left, corresponding to the column indices in the simulation output. The void region was located at the upper right corner of the slab. This labeling scheme facilitates efficient feature extraction and subsequent model development as shown in Figure 7.

#### 2.2.3. Response Extraction and Loading Conditions

To simulate aircraft-induced dynamic responses, vertical (Z-direction) acceleration histories at the bottom of the slab were extracted using the History Output function in ABAQUS as shown in Figure 8. The total simulation time was 0.9 s, consisting of 0.6 s of loading followed by 0.3 s of free decay. A time step of 0.0001 s was used to ensure high-resolution capture of transient dynamic behavior. The resulting data were exported in CSV format for post-processing and feature extraction.

The moving load was modeled based on the 46 × 17R20 tire used in the main landing gear of the Airbus A320. The tire was treated as a rigid body composed of a homogeneous hyperelastic material, with material parameters listed in Table 4. A three-dimensional geometry was created using sweep modeling along a neutral axis and discretized with uniformly distributed C3D8R elements in both circumferential and radial directions, ensuring geometric symmetry and computational accuracy.

Unlike conventional vibration-based SHM approaches that rely on global or manually selected features, this study adopts a data-driven, correlation-guided feature selection strategy, identifying specific sensor–feature pairs with the strongest response to void size. This ensures that model inputs are both highly sensitive and spatially localized, which enhances prediction accuracy and interpretability.

Furthermore, the combination of frequency-domain indicators and physical structural parameters creates a hybrid input space that reflects both dynamic response characteristics and material–geometric conditions, allowing the model to better adapt to varying pavement designs.

## 3. Results and Discussion

### 3.1. Applicability of Vibration Response for Qualitative Evaluation of Sub-Slab Voids

Although modal analysis is commonly used for structural health monitoring, it primarily captures global resonance frequencies under linear assumptions. In contrast, the presence of sub-slab voids introduces nonlinear behavior such as interface separation and contact loss. These effects, especially under moving loads, result in transient responses with energy spread across multiple frequency bands. Therefore, power spectral density (PSD) analysis of transient vibration signals is adopted to capture these dynamic features and quantify the severity of corner voids. To examine the influence of varying void sizes on the distribution of structural vibration energy, this study analyzes the frequency-domain characteristics of the vibration response of Sensor #2 under four void conditions (0 cm, 5 cm, 10 cm, and 15 cm) using the Power Spectral Density (PSD) method. The objective is to evaluate the sensitivity and applicability of PSD parameters for the quantitative assessment of sub-slab voids. The analysis results are presented in Figure 9.

The PSD results reveal notable variations across the different void scenarios, particularly in the dominant frequency and the mid-to-high frequency ranges. Specifically, the 5 cm void condition produces the highest PSD peak, reaching 4.717 × 10^−3^ m^2^/s^4^/Hz at approximately 950 Hz, which is substantially higher than the peak responses observed in other cases. In contrast, the intact slab (0 cm void) exhibits a dominant frequency near 930 Hz with a peak of only 9.14 × 10^−4^ m^2^/s^4^/Hz. For the 10 cm and 15 cm voids, dominant frequencies are approximately 930 Hz and 920 Hz, with corresponding PSD peaks of 8.65 × 10^−4^ and 1.14 × 10^−3^ m^2^/s^4^/Hz, respectively. These results suggest that small-scale voiding (5 cm) induces an initial stiffness degradation, promoting localized modal excitation and more efficient energy activation near the dominant frequency.

An integrated PSD energy analysis within the 0–3000 Hz range reinforces this trend. The 5 cm void condition yields the highest total spectral energy (0.1603 m^2^/s^4^), markedly exceeding the values observed for the 0 cm (0.0467), 10 cm (0.0514), and 15 cm (0.0464) cases. This indicates that small-to-moderate voids enhance focused vibration energy propagation. As the void size increases, however, the total spectral energy diminishes, which is likely due to the slab exhibiting a “cantilever-like” behavior under larger voids. This behavior degrades boundary support and accelerates energy dissipation, particularly in higher vibration modes.

Within the primary frequency band (800–1100 Hz), the 5 cm void condition exhibits the most concentrated and pronounced energy distribution again, with a distinct peak that reflects a transitional phase in stiffness where the structure is most responsive to damage. In comparison, the PSD curves for the 10 cm and 15 cm voids become broader and flatter, indicating modal splitting and energy dispersion as void severity increases, thereby reducing energy concentration in the frequency domain. These findings underscore the significant impact of void growth on modal behavior and energy distribution.

At higher frequencies (above 2000 Hz), the PSD amplitude under the 15 cm void condition exhibits the steepest decline, suggesting considerable energy attenuation. This implies that high-frequency components are particularly susceptible to damping in extensively voided regions, possibly due to the reduced interface stiffness and increased localized damping effects resulting from large-scale delamination.

In summary, PSD analysis not only elucidates the energy excitation patterns under varying void sizes but also demonstrates the high sensitivity of frequency-domain responses to small- and medium-scale voiding. Parameters such as the dominant-frequency PSD peak and integrated spectral energy exhibit strong potential as indicators for real-time sub-slab void detection. Notably, the 5 cm void case reveals marked energy concentration and frequency sensitivity, affirming that power spectral density analysis offers a viable and effective approach for the real-time monitoring and quantitative evaluation of voids beneath concrete pavement slabs.

### 3.2. Time-Frequency Feature Analysis and Correlation Study

#### 3.2.1. Extraction of Time-Domain Vibration Features

Under aircraft wheel impact loading, airport pavement structures often exhibit pronounced nonlinear behavior and localized abrupt responses, producing complex vibration signals with high noise levels that are not directly suitable for void detection. To address this, statistical feature extraction is commonly employed on the time-domain signals, facilitating the construction of feature vectors that characterize structural response patterns.

Time-domain features are typically categorized into two types: dimensional features (e.g., mean, root mean square) and dimensionless features (e.g., kurtosis, skewness, crest factor). Dimensional indicators reflect the signal’s amplitude and energy content, whereas dimensionless indicators are more sensitive to localized anomalies and transient events, which are key manifestations of structural degradation. A summary of commonly used time-domain vibration features is provided in Table 5.

#### 3.2.2. Extraction of Frequency-Domain Vibration Features

To characterize the frequency response under sub-slab void conditions, this study employs Fast Fourier Transform (FFT) to analyze time-domain signals acquired from sensors. Compared to the traditional Discrete Fourier Transform (DFT), FFT offers significantly faster computation with lower computational cost, making it widely applicable in engineering practice. In this context, FFT is used to convert vibration responses into the frequency domain and extract spectral features indicative of structural conditions.

Analogous to time-domain analysis, frequency-domain analysis utilizes statistical metrics to describe structural behavior along the frequency axis. The key features adopted in this study are summarized in Table 6. Among them, feature d_1_ represents the total spectral energy, serving as a measure of global excitation intensity. Features d_2_–d_4_, d_6_, and d_10_–d_13_ describe the distribution of vibrational energy across frequencies, capturing patterns of concentration or dispersion. Features such as d_5_ (spectral centroid) and d_7_–d_9_ (RMS frequency, spectral skewness, and spectral kurtosis) quantify the dominant frequency content and its statistical characteristics.

#### 3.2.3. Correlation Analysis Based on Time–Frequency Domain Features

To investigate the relationship between vibration characteristics and void size, 15 statistical time-domain features—including mean, variance, standard deviation, maximum, minimum, peak, RMS, absolute mean, square root amplitude, waveform index, crest factor, impulse factor, margin factor, skewness (t_14_), and kurtosis—were computed and correlated with void size using Pearson correlation coefficients. The resulting correlation heatmap is presented in Figure 10.

The results reveal that several features—particularly t_14_ (skewness), t_5_ (minimum), t_2_ (variance), t_3_ (standard deviation), and t_11_ (crest factor)—exhibit high absolute correlation values at multiple measure points, indicating strong sensitivity to structural variations induced by slab corner voiding. For example, t_14_ shows the highest correlation (−0.768) at point 2, underscoring its effectiveness in capturing waveform asymmetry resulting from delamination. Similarly, t_2_ and t_7_ (RMS) achieve correlation values exceeding −0.70 at point 0 and 4, reflecting reduced signal energy and variability with increasing void severity. Peak-related features such as t_6_ (peak value) and t_11_ (crest factor) indicate amplified local vibration, likely due to increased flexibility at void edges.

From a spatial perspective, points 0–4, located along the loading path and adjacent to the void region, demonstrate the strongest correlations due to the combined effects of high impact force and localized stiffness degradation. In contrast, points 21–23, although positioned near slab corners, are located above structurally intact regions and exhibit weak correlations, serving as baseline references. Sensors farther from the loading path display relatively stable responses and low correlations, emphasizing the spatial localization of void-induced dynamic changes.

To further assess the impact of voiding on frequency response, 13 spectral features (d_1_–d_13_) were extracted and analyzed. Figure 11 presents the Pearson correlation heatmap between these frequency-domain features and void sizes across all sensor locations. Features such as d_5_ (spectral centroid), d_6_ (frequency standard deviation), and d_10_ (normalized bandwidth) exhibit strong negative correlations (|r| > 0.7) at multiple measure points, indicating a downward shift in dominant frequencies and increased spectral dispersion due to stiffness degradation.

Feature d_5_, in particular, demonstrates a consistent decrease at points 2–4, reflecting the reduction in modal frequencies caused by void-induced softening. Higher-order features such as d_11_ (spectral skewness) and d_12_ (spectral kurtosis) also show pronounced responses, especially at points 3, 22, and 23, highlighting asymmetric energy distribution and localized excitation. d_13_ (mean absolute frequency deviation), a robust metric of frequency dispersion, is notably responsive at boundary sensors, indicating a resilience to outliers and suitability for non-stationary excitation scenarios.

Spatially, points 0–5—aligned with the load path and located near void edges—exhibit the most significant spectral responses, particularly in d_5_, d_6_, and d_10_. Conversely, points 21–23, despite their edge positions, lie above intact substructures and show relatively minor frequency shifts. Sensors located in the central slab region display minimal variation. These findings underscore the spatial selectivity of void effects on frequency-domain features, with the most pronounced impacts observed in regions where loading and voiding coincide.

Overall, these results provide important insights for feature selection and sensor deployment in void detection systems. Sensor–feature pairs exhibiting strong correlations are well-suited as model input variables, while low-response regions may serve as reference baselines or be used for noise suppression in multi-sensor fusion strategies.

It is acknowledged that the correlation analysis is based on a limited number of experimental and simulated cases. Although strong and consistent trends were observed across the two datasets, future work will aim to expand the dataset and apply statistical significance testing to enhance the reliability of feature selection and ranking.

#### 3.2.4. Ranking of Feature Importance Across Sensor Locations

To identify the most representative vibration features for sub-slab void detection, this section constructs a comprehensive feature set comprising 15 time-domain and 13 frequency-domain parameters extracted from each sensor location. From this set, the top 30 sensor–feature pairs with the highest absolute Pearson correlation coefficients with void size were selected, as summarized in Table 7.

The results indicate that the most strongly correlated features are primarily concentrated in several time-domain indicators, such as skewness (t_14_), minimum value (t_5_), variance (t_2_), and peak value (t_3_), as well as frequency-domain indicators such as frequency standard deviation (d_6_) and normalized bandwidth (d_10_). Notably, the t_14_–P2 combination exhibits the highest correlation coefficient (−0.768), highlighting its high sensitivity to waveform asymmetry induced by void formation. Frequency-related features, particularly d_6_ and d_10_, also demonstrate strong correlations (>0.70) at sensor locations P0 and P3, reflecting the pronounced impact of bandwidth broadening and modal frequency shifts associated with structural stiffness degradation.

In terms of spatial distribution, the most sensitive point–feature combinations are predominantly located at P0, P2, P3, and P4—positions near the slab corner intersecting the wheel load path—where the interaction between impact excitation and local stiffness variation is most prominent. In contrast, sensors positioned at the slab center or in intact (non-voided) regions are absent from the top 30 combinations, further confirming the spatial localization of vibration features in response to voiding.

Additionally, correlation analysis between the normalized signal energy (W_n_) and void size at key sensor locations yielded high coefficients: −0.911 (P0), −0.934 (P2), −0.824 (P3), and −0.815 (P4). These results provide strong evidence for the effectiveness of targeted point–feature selection in predictive modeling and support the superiority of localized feature extraction over global aggregation strategies. This approach reduces the inclusion of redundant or irrelevant dimensions that could otherwise degrade model performance.

### 3.3. Void Identification Index Construction and Model Design

#### 3.3.1. Design of Input and Output Variables

Based on the preceding correlation analysis between point–feature combinations and void size, this section selects the normalized power spectral density energy features (W_n_) from four representative sensor locations (P0, P2, P3, and P4) as the core input variables for model development. To further improve the model’s adaptability to structural variations, five structural parameters are incorporated as supplementary inputs: surface layer thickness, base layer thickness, elastic modulus of the surface layer, elastic modulus of the base layer, and elastic modulus of the subgrade. The final input feature matrix thus comprises four high-correlation spectral energy indicators and five structural property variables.

The output variable of the model is the void size corresponding to each sensor configuration.

To eliminate the effects of differing variable scales and to ensure a balanced model training, all input variables are standardized using Z-score normalization, resulting in features with zero mean and unit variance. The dataset is then randomly divided into a training set (75%) and a testing set (25%), ensuring an adequate number of samples for model training while retaining independent data for performance evaluation. The random seed for splitting the training and testing data was fixed at 42 to ensure the reproducibility of the modeling results.

The final input matrix includes vibration indicators with high correlation (W_n_) extracted from specific sensor–feature combinations, as well as five structural parameters of the pavement. All input variables were standardized using Z-score normalization. The output is the known void size. The dataset was divided into training and testing sets with a 75:25 split.

#### 3.3.2. Model Construction and Hyperparameter Tuning

To account for potential nonlinear relationships between input features and void size, as well as the coupling of multi-source inputs and the inherent uncertainty in sensor signals, a Random Forest Regression model is adopted as the primary predictive framework. Random Forest is a non-parametric ensemble learning method based on the Bagging strategy, which aggregates the outputs of multiple independently trained decision trees by averaging their predictions. Compared to traditional linear models, Random Forests offer superior capabilities in capturing complex nonlinear mappings, handling high-dimensional input spaces, and exhibiting robustness to outliers, making them particularly well-suited for structural health monitoring applications involving multi-source feature fusion and vibration-based response prediction.

To optimize the model’s predictive performance, a grid search algorithm combined with 5-fold cross-validation is employed for systematic hyperparameter tuning. The tuning process targets key hyperparameters, including the maximum tree depth (max_depth), the minimum number of samples required at a leaf node (min_samples_leaf), and the total number of trees in the forest (n_estimators). The optimal hyperparameter configuration identified through this process is summarized in Table 8.

The Random Forest model was trained using grid search to optimize hyperparameters such as the number of trees and maximum depth. A 5-fold cross-validation strategy was employed on the training set to avoid overfitting. The performance was evaluated using three metrics: R^2^, RMSE, and MAE, computed on the test set.

#### 3.3.3. Prediction Performance and Error Analysis

To evaluate the performance of the proposed Random Forest (RF) model in the quantitative estimation of sub-slab void size, three commonly used regression metrics were employed: the coefficient of determination (R^2^), root mean square error (RMSE), and mean absolute error (MAE). Using the previously constructed feature matrix and optimized hyperparameter settings, the RF model was trained and tested. The model exhibited excellent predictive performance on the test set, achieving R^2^ = 0.991, RMSE = 0.054 m, and MAE = 0.021 m, indicating that it could explain 99.1% of the variance in the target variable while maintaining prediction errors within approximately 6 cm. This level of accuracy demonstrates the model’s effectiveness in detecting and quantitatively assessing corner voids in pavement structures.

Figure 12 presents a comparison between predicted and actual void sizes in the test dataset. The horizontal axis denotes the true void size, while the vertical axis represents the predicted values. The dashed reference line (y = x) indicates perfect prediction. The predicted points closely cluster around the reference line, reflecting a high consistency and strong model fit across the entire range of void sizes. In the small-to-moderate void range (0–10 cm), the predictions are nearly indistinguishable from actual values, with minimal error. Even at larger void sizes (e.g., 15 cm), the model maintains excellent tracking accuracy, showing no significant underestimation, overestimation, or systematic deviation. The absence of strip-like dispersion or biased clustering patterns further confirms that the model avoids overfitting and delivers stable, generalizable predictions.

Figure 13 illustrates the probability distribution of prediction errors, offering insight into the statistical characteristics and bias behavior of the residuals. The histogram displays the frequency of error samples across defined intervals, overlaid with a kernel density estimate (KDE) curve to indicate the continuous distribution. The error distribution exhibits a near-zero-centered, symmetric, and unimodal shape, with the majority of residuals falling within the range of [−0.05, +0.05] m. The peak of the distribution is located near zero, confirming the absence of systematic bias or skewness. Additionally, the steep attenuation of the distribution tails suggests a lack of long-tailed or extreme error values, demonstrating the model’s robustness and resistance to outliers, even under limited data conditions.

Taken together, Figure 12 and Figure 13 provide compelling evidence—both in terms of predictive consistency and residual statistics—that the Random Forest model, developed using high-correlation W_n_ features and structural parameters, performs exceptionally well in predicting void size. The model not only delivers accurate estimates across the full void range but also maintains excellent control over prediction errors. The results indicate that the RF model is capable of accurately predicting void size from hybrid input features, confirming its suitability for capturing the complex, nonlinear relationships present in vibration response data.

While the Random Forest model performs well in terms of accuracy and residual control, future work will include comparative evaluation with alternative modeling strategies, such as SVM, gradient boosting methods, or deep neural networks. These comparisons will help quantify trade-offs in interpretability, complexity, and generalization under varying input features. Additionally, efforts will be made to extend validation using in situ pavement monitoring datasets to better assess model transferability and real-world readiness.

## 4. Conclusions

This study presents a comprehensive methodology for detecting and quantifying sub-slab voids in airport concrete pavements based on vibration responses under moving load conditions. The major conclusions are summarized as follows:The combination of scaled indoor impact testing and finite element simulation under realistic moving loads effectively captures the vibration response characteristics associated with various void sizes, particularly at vulnerable slab corners.Power spectral density (PSD), time-domain statistics (e.g., skewness, crest factor), and frequency-domain metrics (e.g., spectral centroid, bandwidth) exhibit high sensitivity to void-induced stiffness degradation. Feature–location correlation analysis revealed that sensor data near the void boundary best reflect void severity.Based on the constructed simulation dataset, the RF model achieved strong fitting performance in estimating corner void size, showing clear relevance between spectral indicators and geometric characteristics. These findings underscore the promising potential of vibration-based machine learning approaches for structural health assessment in rigid pavement systems.

With the development of mobile sensing platforms, aircraft-mounted accelerometers, and wireless sensor networks, the proposed vibration-based framework is extendable to full-scale airport pavement monitoring. Response data can be collected during regular aircraft taxiing or mobile inspections without disrupting operations, enabling efficient and scalable void detection across large areas. Future work will focus on extending the proposed method to full-scale field validation campaigns under in-service aircraft loading, and accounting for material aging, construction variability, and structural discontinuities to improve generalizability.

## Figures and Tables

**Figure 1 sensors-25-04703-f001:**
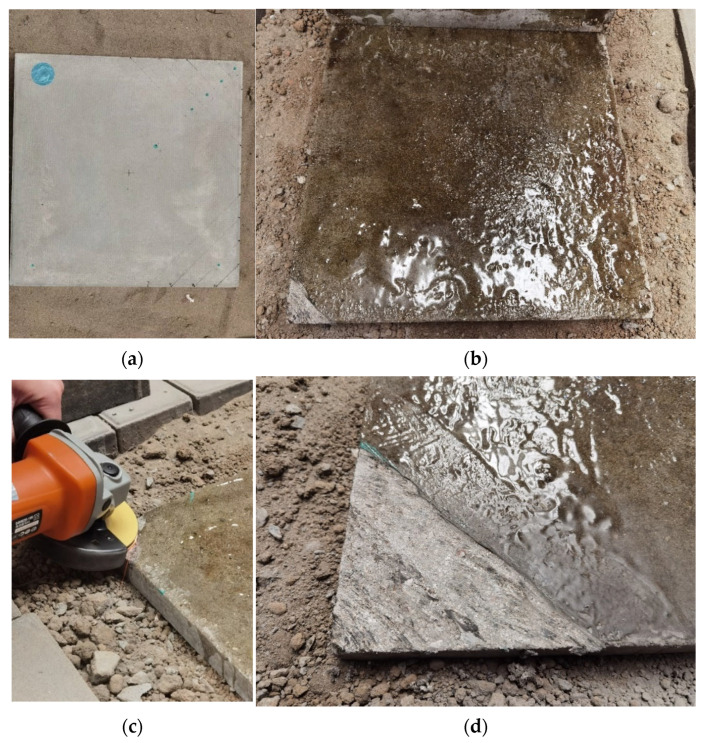
Preliminary treatment prior to the scaled model test: (**a**) Specimen preparation; (**b**) Bonding treatment between the surface and base courses; (**c**) Angle grinder treatment; (**d**) Artificial void treatment.

**Figure 2 sensors-25-04703-f002:**
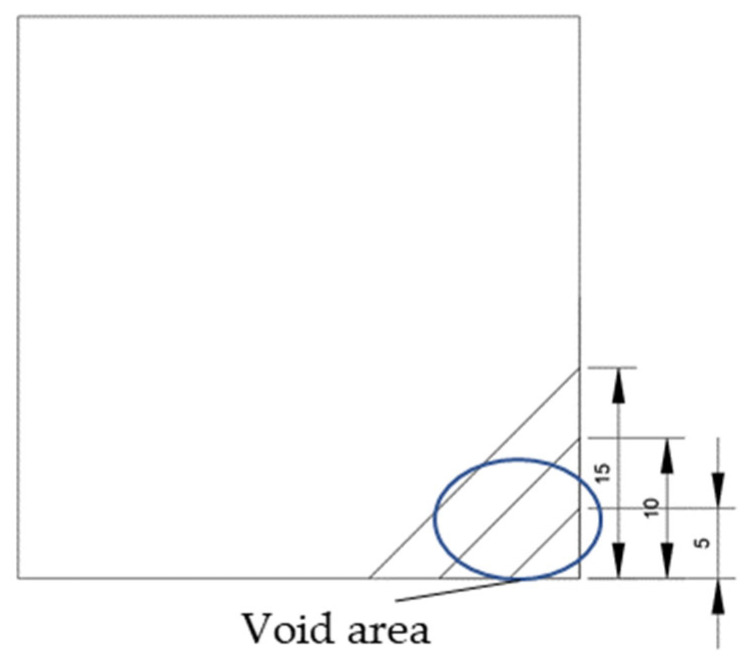
Scaled test slab with simulated corner void.

**Figure 3 sensors-25-04703-f003:**
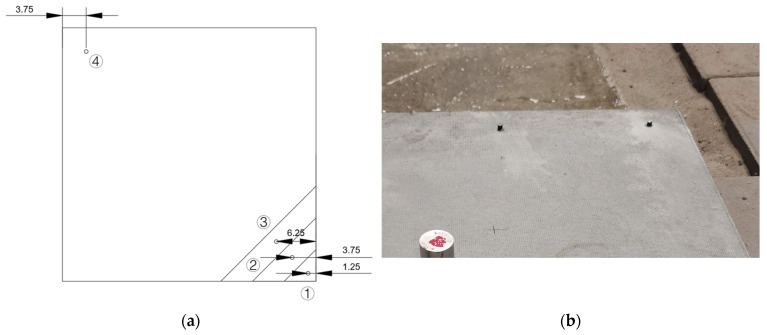
Sensor layout and installation: (**a**) Placement of sensors; (**b**) Installation of sensors.

**Figure 4 sensors-25-04703-f004:**
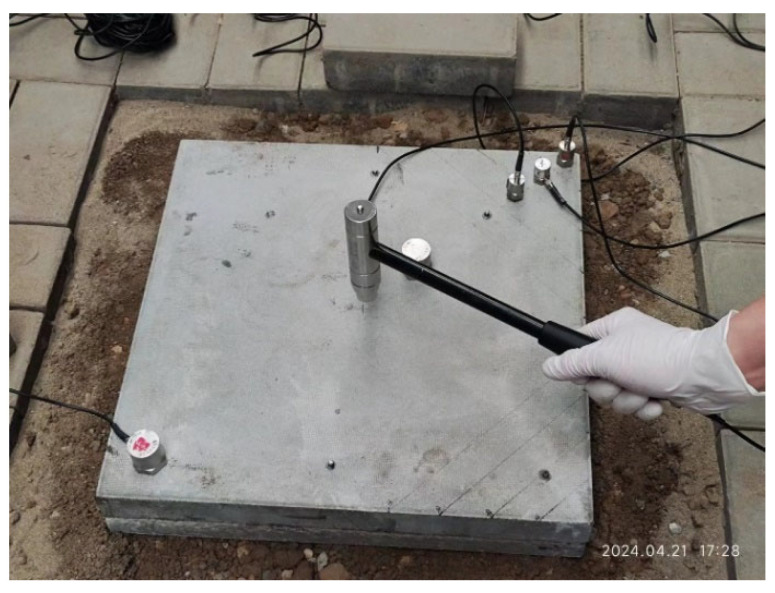
Loading method.

**Figure 5 sensors-25-04703-f005:**
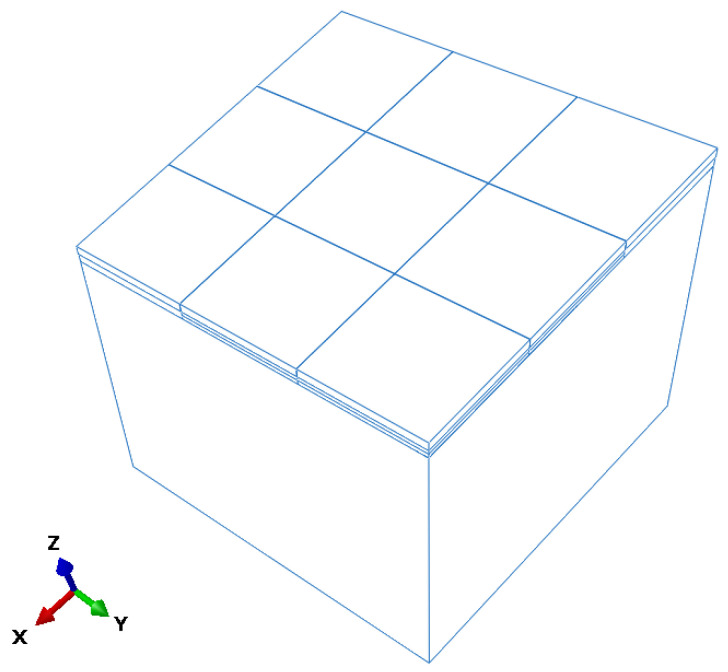
Three-dimensional geometry of the simulation model of the pavement slab.

**Figure 6 sensors-25-04703-f006:**
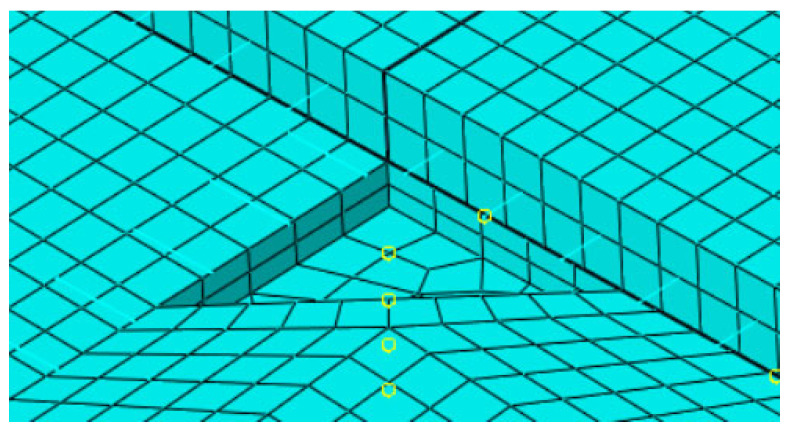
Finite element model of the void region.

**Figure 7 sensors-25-04703-f007:**
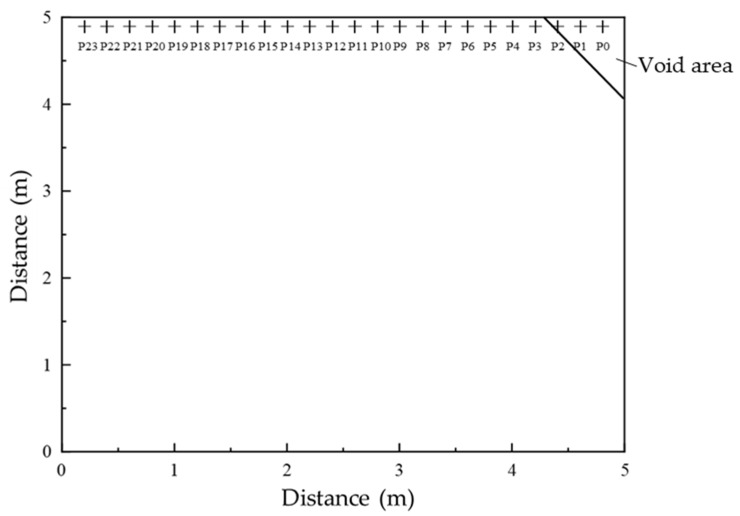
Distribution and numbering of vibration response measurement points.

**Figure 8 sensors-25-04703-f008:**
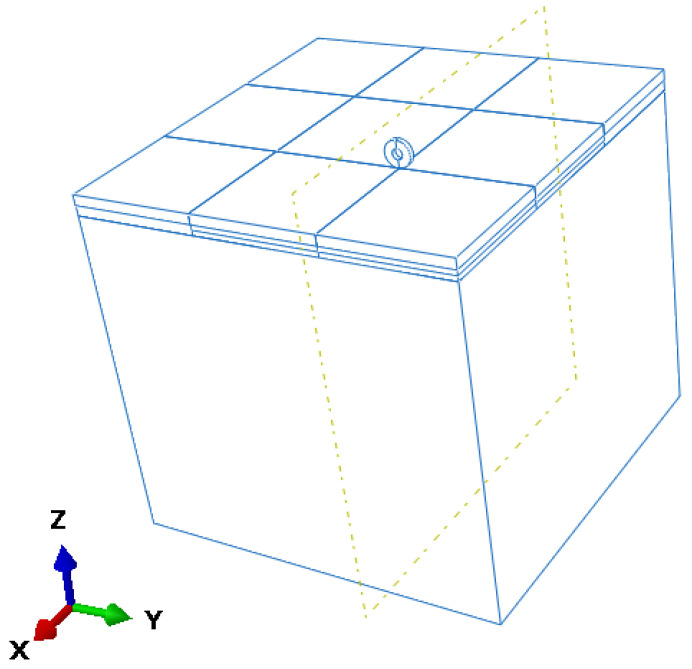
Three-dimensional geometry of the refined simulation model with sub-slab void.

**Figure 9 sensors-25-04703-f009:**
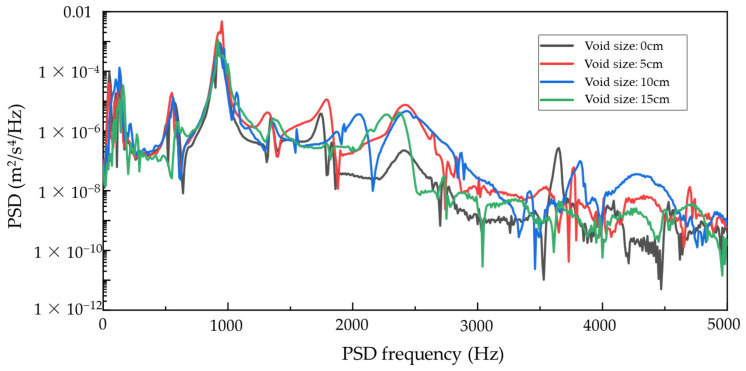
PSD response of Sensor #2 under varying void conditions (central impact).

**Figure 10 sensors-25-04703-f010:**
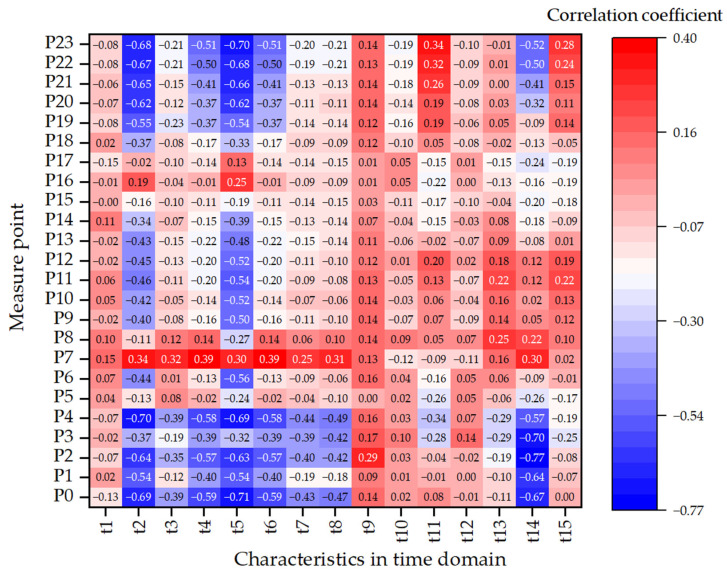
Heatmap illustrating the correlation between time-domain vibration features and void size across all sensor locations.

**Figure 11 sensors-25-04703-f011:**
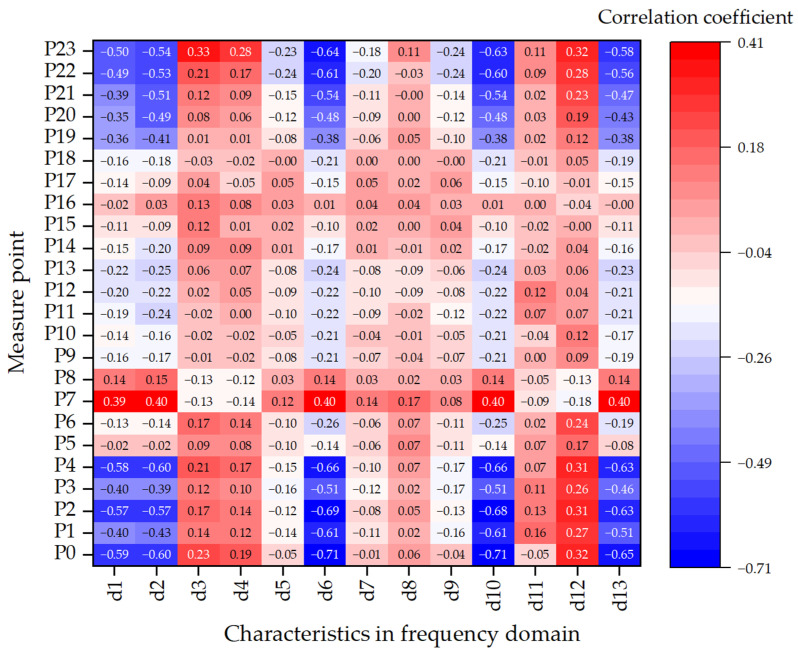
Heatmap illustrating the correlation between frequency-domain vibration features and void size across all sensor locations.

**Figure 12 sensors-25-04703-f012:**
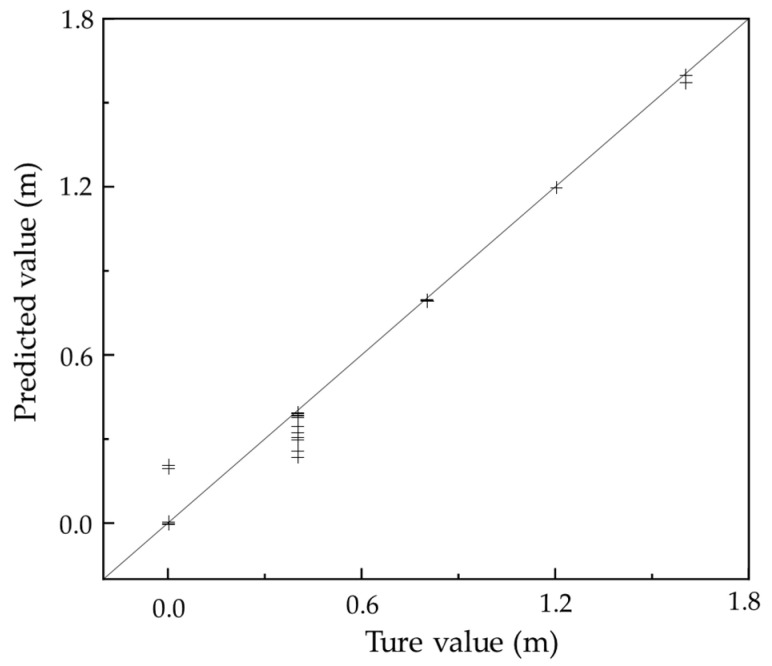
Scatter plot of true and predicted void sizes.

**Figure 13 sensors-25-04703-f013:**
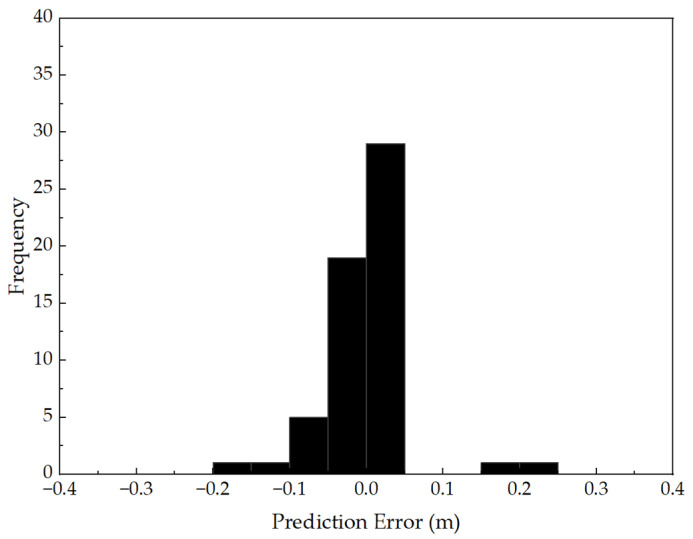
Distribution of Prediction Errors in Void Size Estimation.

**Table 1 sensors-25-04703-t001:** Geometry and performance characteristics of the piezoelectric accelerometer.

Type	INV9821	INV9822	INV9823	INV9824	INV9828
Sensitivity (mV/g)	50	100	200	5	500
Range (g)	100	50	25	1000	10
Resolution ratio (m/s^2^)	0.001	0.0005	0.00025	0.01	0.0001
Frequency response (Hz)	0.5~5 k	0.5~8 k	0.5~4 k	1~15 k	0.2~2.5 k
Resonant frequency (Hz)	≥25 k	≥25 k	≥8 k	40 k	8 k
Operating temperature (°C)	−40~+120	−40~+120	−40~+120	−40~+120	−40~+120
Maximum impact (g)	2000	2000	1000	3000	1000
Operating voltage (v)	+18~+28	+18~+28	+18~+28	+18~+28	+18~+28
Operating current (mA)	2~20	2~20	2~20	2~20	2~20
Output impedance (Ω)	<100	<100	<100	<100	<100

**Table 2 sensors-25-04703-t002:** Material parameters of the pavement model.

Structural Materials	Elasticity Modulus (Mpa)	Dynamic Modulus of Elasticity (MPa)	Density (kg/m^3^)	Poisson’s Ratio	Damping Coefficient α (s^−1^)	Damping Coefficient β (s^−1^)
Cement concrete surface layer	36,000	49,819	2500	0.15	5	0.003
Cement-stabilized base layer	1500	2692	2000	0.25	0.82	0.0122
Subgrade	80	242	1800	0.35	0.42	0.0061
Dowel bar	210,000		7800	0.3	0	0

**Table 3 sensors-25-04703-t003:** Simplified Corner Voiding Condition of Concrete Pavement Slab.

Working condition	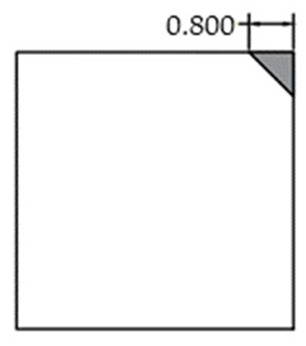	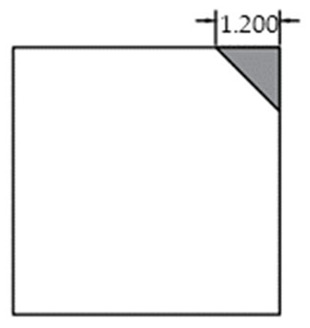	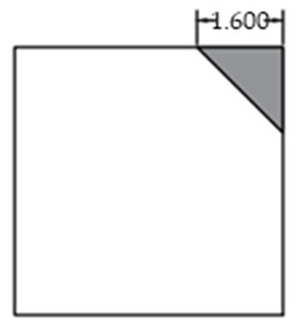	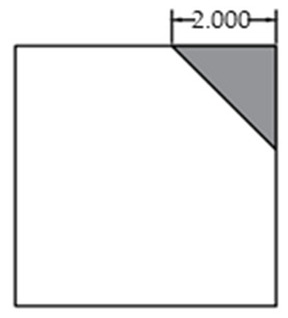
Void size (m)	0.8	1.2	1.6	2

**Table 4 sensors-25-04703-t004:** Tire model parameters.

Tire Model Parameters	Value
Outer diameter of the tire (mm)	1168.4
Inner diameter of the tire (mm)	508
Section width of the tire (mm)	431.8
Positive-definite material parameters C_10_	9.9 × 10^6^
Positive-definite material parameters C_01_	8.8 × 10^6^
Incompressibility parameter D_1_	10^−7^
Rated tire pressure (kPa)	1590

**Table 5 sensors-25-04703-t005:** Time-domain features of pavement vibration signals.

Number	Expression	Number	Expression
1	$t_{1}=\frac{\sum_{n=1}^{N}x(n)}{N}$	9	$t_{9}={\left(\frac{\sum_{n=1}^{N}\sqrt{x\left(n\right)}}{N}\right)}^{2}$
2	$t_{2}=\frac{\sum_{n=1}^{N}{\left(x\left(n\right)-t_{1}\right)}^{2}}{N-1}$	10	$t_{10}=\frac{t_{7}}{\frac{\sum_{n=1}^{N}\left|x(n)\right|}{N}}$
3	$t_{3}=\sqrt{\frac{\sum_{n=1}^{N}{\left(x\left(n\right)-t_{1}\right)}^{2}}{N}}$	11	$t_{11}=\frac{t_{6}}{t_{7}}$
4	$t_{4}=\max x(n)$	12	$t_{12}=\frac{t_{6}}{\frac{\sum_{n=1}^{N}\left|x(n)\right|}{N}}$
5	$t_{5}=\min x(n)$	13	$t_{13}=\frac{t_{6}}{t_{9}}$
6	$t_{6}=\max\left|x(n)\right|$	14	$t_{14}=\frac{\sum_{n=1}^{N}{\left(x\left(n\right)-t_{1}\right)}^{3}}{\left(N-1\right)t_{2}^{3}}$
7	$t_{7}=\sqrt{\frac{\sum_{n=1}^{N}{\left(x\left(n\right)\right)}^{2}}{N}}$	15	$t_{15}=\frac{\sum_{n=1}^{N}{\left(x\left(n\right)-t_{1}\right)}^{4}}{\left(N-1\right)t_{2}^{4}}$
8	$t_{8}=\frac{\sum_{n=1}^{N}\left|x(n)\right|}{N}$		

Note: *x*(*n*) represents the time-domain signal sequence; *n* denotes the index of the sampling point; and *N* is the total number of sampling points.

**Table 6 sensors-25-04703-t006:** Frequency-domain vibration features of pavement slab.

Number	Expression	Number	Expression
1	$d_{1}=\frac{\sum_{k=1}^{K}\;s(k)}{K}$	8	$d_{8}=\sqrt{\frac{\sum_{k=1}^{K}\;{f_{k}}^{4}s(k)}{\sum_{k=1}^{K}\;{f_{k}}^{2}s(k)}}$
2	$d_{2}=\frac{\sum_{k=1}^{K}\;(s(k)-d_{1}{)}^{2}}{K-1}$	9	$d_{9}=\frac{\sum_{k=1}^{K}\;f_{k}^{2}s(k)}{\sqrt{\sum_{k=1}^{K}\;s(k)\sum_{k=1}^{K}\;f_{k}^{4}s(k)}}$
3	$d_{3}=\frac{\sum_{k=1}^{K}\;(s(k)-d_{1}{)}^{3}}{K(\sqrt{d_{2}}{)}^{3}}$	10	$d_{10}=\frac{d_{6}}{d_{5}}$
4	$d_{4}=\frac{\sum_{k=1}^{K}\;(s(k)-d_{1}{)}^{4}}{K{d_{2}}^{2}}$	11	$d_{11}=\frac{\sum_{k=1}^{K}\;(f_{k}-d_{5}{)}^{3}s(k)}{K{d_{6}}^{3}}$
5	$d_{5}=\frac{\sum_{k=1}^{K}\;f_{k}s(k)}{\sum_{k=1}^{K}\;s(k)}$	12	$d_{12}=\frac{\sum_{k=1}^{K}\;(f_{k}-d_{5}{)}^{4}s(k)}{K{d_{6}}^{4}}$
6	$d_{6}=\sqrt{\frac{\sum_{k=1}^{K}\;(f_{k}-d_{5}{)}^{2}s(k)}{K}}$	13	$d_{13}=\frac{\sum_{k=1}^{K}\;(f_{k}-d_{5}{)}^{1/2}s(k)}{K\sqrt{d_{6}}}$
7	$d_{7}=\sqrt{\frac{\sum_{k=1}^{K}\;{f_{k}}^{2}s(k)}{\sum_{k=1}^{K}\;s(k)}}$		

Note: *s*(*k*) denotes the signal spectrum; *k* is the index of the spectral line; *K* represents the total number of spectral lines; and *fₖ* indicates the frequency value corresponding to the *k*-th spectral line.

**Table 7 sensors-25-04703-t007:** Top 30 point–feature combinations with highest correlation to void size.

Feature	Measure Point	Correlation	Type	|Correlation|
$t_{14}$	P2	−0.76785	Time domain	0.767851
$t_{5}$	P0	−0.71114	Time domain	0.711137
$d_{10}$	P0	−0.70755	Frequency domain	0.707551
$d_{6}$	P0	−0.70702	Frequency domain	0.707018
$t_{14}$	P3	−0.70196	Time domain	0.701963
$t_{5}$	P23	−0.69554	Time domain	0.695536
$t_{2}$	P4	−0.69529	Time domain	0.695294
$t_{5}$	P4	−0.69265	Time domain	0.692652
$d_{6}$	P2	−0.68714	Frequency domain	0.68714
$t_{2}$	P0	−0.687	Time domain	0.686997
$t_{5}$	P22	−0.68091	Time domain	0.68091
$d_{10}$	P2	−0.68046	Frequency domain	0.680459
$t_{2}$	P23	−0.67833	Time domain	0.678326
$t_{14}$	P0	−0.67162	Time domain	0.671618
$t_{2}$	P22	−0.66782	Time domain	0.667824
$d_{6}$	P4	−0.66	Frequency domain	0.660003
$d_{10}$	P4	−0.65935	Frequency domain	0.659349
$t_{5}$	P21	−0.65817	Time domain	0.658171
$d_{13}$	P0	−0.65394	Frequency domain	0.653938
$t_{2}$	P21	−0.65096	Time domain	0.650957
$t_{14}$	P1	−0.64009	Time domain	0.640092
$t_{2}$	P2	−0.63841	Time domain	0.638406
$d_{6}$	P23	−0.63816	Frequency domain	0.63816
$d_{10}$	P23	−0.6346	Frequency domain	0.6346
$d_{13}$	P2	−0.6327	Frequency domain	0.632705
$t_{5}$	P2	−0.63014	Time domain	0.630143
$d_{13}$	P4	−0.62525	Frequency domain	0.625254
$t_{2}$	P20	−0.61913	Time domain	0.619132
$t_{5}$	P20	−0.61768	Time domain	0.617682
$d_{6}$	P1	−0.60987	Frequency domain	0.609872

**Table 8 sensors-25-04703-t008:** Random Forest model configuration parameters.

Parameter	Parameter Meaning	Search Range	Value
n_estimators	Number of Estimators	50, 100, 150, 200, 250, 300	250
max_depth	Maximum Tree Depth	5, 10, 15, 20, 25, None	15
min_samples_leaf	Minimum Samples per Leaf Node	1, 2, 3, 4, 5	1
random_state	Random Seed	42	42

## Data Availability

Data is contained within the article.

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
