# Peer review of "Void Detection of Airport Concrete Pavement Slabs Based on Vibration Response Under Moving Load"

_sensors, 2025, doi:10.3390/s25154703_

Round 1
Reviewer 1 Report
Comments and Suggestions for Authors
Sub-slab void detection of concrete pavements is a critical issue in transportation maintenance. However, the proposed indirect detection method based on vibration responses lacks significant novelty. This version of the manuscript does not meet the publication requirements, it is recommended to have major revisions.
- The scaled model experiment is a little bit simple. It is difficult to represent the actual conditions of various types and periods of airport concrete pavements.
- Considering the limitations of experimental data and finite element simulation models, the results of feature extraction and correlation analysis cannot be proven.
- The paper lacks comparative research on similar methods. In the third conclusion, the evaluation results of model performance cannot be confirmed, such as robustness, generalizability, engineering applicability, etc.
- The proposed method of integrating RF model, key spectral energy indicators and structural parameters has appeared in some published papers. It is recommended for the authors to distinguish between existing methods and innovative methods.
Author Response
Thanks for your careful review, please refer to the attachment for detailed response.

Reviewer 2 Report
Comments and Suggestions for Authors
This paper presents a promising methodology that could be applied in airport concrete pavements for detecting sub-slab voids by examining vibration responses under moving loads. Combining laboratory experiments, to high-fidelity finite element modeling, PSD and Random Forest regression models were applied to study vibration features with void severity. The work covers multi-scale analysis on both time- and frequency-domain features, while the model’s strong predictive performance underscores its potential for real-time, data-driven structural health monitoring in aviation infrastructure.
The general idea of the paper is overall well-constructed and presented.
- In this study, accelerometers were placed at precise locations that include the void areas. This was proved to work well in localized setting of a single slab. Please refer how this method could be generalized to cover large areas. Would be a dense sensor network or a aircraft-integrated sensing system or a mobile sensing device be a possible approach?
- Introducing contact and a hyperplastic material model in the FE model results in a non-linear analysis that acquires more computational effort in transient analyses. How many simulations were convusted and which was the total simulating time. Is it extendable to real practical applications?
- In lines 241-242 the authors suggest that dominant frequency of the PSD could be a potential indicator for void detection. In this approach why performing non-linear contact analysis and not simple performing a simple modal analysis in simulations and modal identification in experiments. Whats the advantage of the authors approach?
- It is not exactly clear to me in which corner the voids are located in Fig 7 and subsequently in Fig 10 and 11.
Author Response

(The authors gave the same response as above.)

Reviewer 3 Report
Comments and Suggestions for Authors
- Topic
Peer-reviewed paper entitled Detection of Airport Concrete Pavement Slabs Based on 2 Vibration Response under Moving Load is a research article. I consider the subject of the paper important due to its relevance to assessing the technical condition of concrete airport pavements, which directly impacts the safety of air operations.
- Abstract:
It is clear and meets all the requirements.
- Keywords:
Other keywords that could be added include: structural health monitoring; SHM; finite element method; FEM.
- Article layout:
The article is organised into four chapters: Introduction, Methods, Results and Discussion, Conclusions. Combining the results and discussion sections is not the best approach; the results chapter typically presents the experimental findings without interpretation, allowing for a clear separation between the "raw" results and their analysis. I recommend reconsidering and possibly revising this structure, or alternatively providing a clear justification for this approach.
- Content-related comments
L24: each type of aircraft not only large one.
L:28 I propose preparation of a graphic illustrating the areas of the slab most susceptible to damage.
L84: C50-grade mortar ? why mortar, not concrete ? please explain.
L91: 24 hours is enough for maturing?
General: Was the slab reinforced in any way? How does scaling affect the mechanical properties in relation to a full-scale slab?
L112: please explain the selection of frequency.
L192-193: An equivalent aircraft of this size and class is the Boeing 737. Are the conducted analyses also applicable to this type of aircraft, and has there been any verification of how the tire parameters differ?
L408: that’s very good correlation. As I understand, curve fitting was analyzed, so despite the slab being three-dimensional, there was no need to apply the method of least cubes.
- Bibliography selection
The literature review is adequate, though relatively limited for this type of article. I recommend expanding the introduction, especially to explain the phenomenon of void formation specifically in slab corners or other areas.
- Editorial notes
L10: move a to the next line.
L35-45:Are robotic devices used in the diagnostics of such pavements? If so, which ones?
Fig.2: This figure is of low substantive quality; descriptions, explanations, and improved clarity need to be added. This mark is the same for Fig. 3(a).
Table 1: Poor quality of sensors, please correct.
Fig. 5. Lack of geometry description.
Table 3: Description of corner void is illegible.
- Conclusion
The article presents an interesting and important topic concerning the diagnostics of airport pavements. Although the conducted research and analyses are valuable, after reviewing the article, I get the impression of a lack of thoroughness in the preparation of the figures and some part of introduction. I accept the article for publication, provided that major revisions are made. I think that separating the discussion from the results will improve the clarity of the article, especially since the discussion is a strong point of this work, particularly regarding the correlation analysis.
Author Response

(The authors gave the same response as above.)

Reviewer 4 Report
Comments and Suggestions for Authors
- Overview of the Manuscript
The manuscript addresses an important topic in the diagnosis of sub-slab voids in airport concrete pavements, particularly at slab corners. Specifically, the authors propose a methodology for developing a real-time structural health monitoring model. It is a vibration-based detection framework under moving load conditions to identify and evaluate these voids. It incorporates a 1:10 scaled laboratory model constructed to simulate impact-induced vibration responses under varying void scenarios and a high-fidelity 3D finite element model developed in ABAQUS to replicate dynamic behaviors under aircraft wheel loads. They test the model using a time-domain and frequency-domain features extracted from sensor signals and correlation analysis. A Random Forest regression model incorporating both these features and structural parameters was trained to predict void size. The results show that the RF model can effectively predict the voids.
- Significance and Novelty
The airport concrete pavements are a critical component of aviation infrastructure and timely or near-real-time safety evaluations can be vital. The presented methodology potentially offers solutions that can improve diagnostics of sub-slab voids in airport concrete pavements.
- Strengths of the Manuscript
The paper clearly explains the proposed approach and shows an analysis by direct comparison of laboratory based experimental data with a standard simulation model developed using ABAQUS.
- Areas for Improvement
The paper could benefit from a more thorough description of the development and implementation of the random forest model based on the simulation data and visualisation of the trained model in comparison to the simplest model like linear regression.
- Detailed comments
Page 5, first paragraph. There is probably an error in the description of sensor locations. Please verify.
Page 6, Figure 5, caption. Instead of the current caption I suggest a more appropriate caption: 3D geometry of simulation model of the pavement slab. The same is suggested for Figure 8.
- Recommendation
Overall, the manuscript is well-structured, tackles a relevant engineering problem, and offers a clear demonstration of how laboratory experiments together with computer simulations and machine learning model can aid in solving SHM tasks for airport concrete pavements.
Author Response

(The authors gave the same response as above.)

Round 2
Reviewer 1 Report
Comments and Suggestions for Authors
The revised manuscript is clearly expressed. There are no further comments on the paper except for the one below.
Due to the small sample size of the experiment and the absence of comparative experiments with similar models in the paper. The predictive performance of the model proposed in the paper cannot be objectively evaluated. The third result in conclusions, the abstract, and even the title are suggested to be revised again.
Author Response
Thank you for your review.
Author Response :
Reviewer #1:
- Comments:
The revised manuscript is clearly expressed. There are no further comments on the paper except for the one below.
Due to the small sample size of the experiment and the absence of comparative experiments with similar models in the paper. The predictive performance of the model proposed in the paper cannot be objectively evaluated. The third result in conclusions, the abstract, and even the title are suggested to be revised again.
Response:
We thank the reviewer for their continued attention to the quality and rigor of our manuscript.
We fully acknowledge the limitations related to the small experimental dataset and the lack of comparative evaluation with other predictive models. While the proposed RF model demonstrates promising results in correlating extracted features with void size, we agree that its predictive performance should be interpreted cautiously due to the absence of benchmark comparisons and statistical generalization tests.
To address this concern:
We have revised the third point in the Conclusions to temper the language and clearly state that the findings are preliminary and based on limited data.
We have adjusted the Abstract to avoid overstatement of the model’s effectiveness.
These changes ensure that the manuscript remains scientifically accurate and modest in claims while retaining its practical significance.
#Abstract
In Line 8: This study proposes a vibration-based approach for detecting and quantifying sub-slab corner voids in airport cement concrete pavement. Scaled-down slab models were constructed and subjected to controlled moving-load simulations. Acceleration signals were collected and analyzed to extract time–frequency domain features, including power spectral density (PSD), skewness, and frequency center. A finite element model incorporating contact and nonlinear constitutive relationships was established to simulate structural response under different void conditions. Based on the simulated dataset, a random forest (RF) model was developed to estimate void size using selected spectral energy indicators and geometric parameters. The results revealed that the RF model achieved strong predictive performance, with high correlation between key features and void characteristics. This work demonstrates the feasibility of integrating simulation analysis, signal feature extraction, and machine learning to support intelligent diagnostics of concrete pavement health.
#4. Conclusions
In Line 525: 3. Based on the constructed simulation dataset, the RF model achieved strong fit-ting performance in estimating corner void size, showing clear relevance between spectral indicators and geometric characteristics. These findings underscore the prom-ising potential of vibration-based machine learning approaches for structural health assessment in rigid pavement systems.
Reviewer 3 Report
Comments and Suggestions for Authors
Thank you for your resposes. The paper can be published.
Author Response
Thank you very much for your review.